# Joint Similarity Item Exploration and Overlapped User Guidance for Multi-Modal Cross-Domain Recommendation

## ABSTRACT

Cross-Domain Recommendation (CDR) has been widely investigated for solving long-standing data sparsity problem via knowledge sharing across domains. In this paper, we focus on the Multi-Modal Cross-Domain Recommendation (MMCDR) problem where different items have multi-modal information while few users are overlapped across domains. MMCDR is particularly challenging in two aspects: fully exploiting diverse multi-modal information within each domain and leveraging useful knowledge transfer across domains. However, previous methods fail to cluster items with similar characteristics while filtering out inherit noises within different modalities, hurdling the model performance. What is worse, conventional CDR models primarily rely on overlapped users for domain adaptation, making them ill-equipped to handle scenarios where the majority of users are non-overlapped. To fill this gap, we propose Joint Similarity Item Exploration and Overlapped User Guidance (**SIEOUG**) for solving the MMCDR problem. **SIEOUG** first proposes similarity item exploration module, which not only obtains pair-wise and group-wise item-item graph knowledge, but also reduces irrelevant noise for multi-modal modeling. Then **SIEOUG** proposes user-item collaborative filtering module to aggregate user/item embeddings with the attention mechanism for collaborative filtering. Finally **SIEOUG** proposes overlapped user guidance module with optimal user matching for knowledge sharing across domains. Our empirical study on Amazon dataset with several different tasks demonstrates that **SIEOUG** significantly outperforms the state-of-the-art models under the MMCDR setting.

## CCS CONCEPTS

• **Information systems → Recommender systems**.

## KEYWORDS

Recommendation, Cross-Domain Recommendation, Multi-Modal Cross-Domain Recommendation, Domain Adaptation

**ACM Reference Format:**
Anonymous Author(s). 2023. Joint Similarity Item Exploration and Overlapped User Guidance for Multi-Modal Cross-Domain Recommendation. In *The Web Conference(WWW '25), April 28 – May 2, 2025, Sydney, Australia.* ACM, New York, NY, USA, 11 pages.

## 1 INTRODUCTION

Recommender systems become more and more attractive with the big data explosion in recent years [22, 38, 48]. Users may participate in multiple domains, e.g., picking fashion clothes in *Ebay* and buying sport outlets in *Amazon* platforms. Meanwhile items on different domains also inherit useful diverse multi-modal visual or text information. However, users may have limited rating interactions across different domains, leading to data sparsity issues in each domain. Thus how to enhance model performance across domains by leveraging multi-modal information with knowledge sharing still needs more investigation.

In this paper, we focus on Multi-Modal Cross-Domain Recommendation (MMCDR) problem as shown in Fig.1. That is, users among different domains are partially overlapped with sparse user-item rating interactions. Meanwhile items on different domains are heterogeneous and they also contain multi-modal information including text and image descriptions. Item multi-modal information is rather valuable since it is extra information reflecting the users tastes beyond rating interactions. For instance, when we discover the user who prefers luxury clothes via multi-modal item information, he/she may also have great interests in fashion sport outlets as shown in Fig.1. There are two main challenges for solving MMCDR: (1) **CH1**: How to better exploit and utilize diverse multi-modal item information within each domain? (2) **CH2**: How to leverage useful knowledge among these partially overlapped users to alleviate data sparsity problem across domains?

However, previous methods cannot better solve MMCDR problem well. On the one hand, previous multi-modal recommendation models [17, 68] mainly focus on exploring item similarity information from a perspective of pair-wise item-item similarity. However, different modalities may inherit noises [68] that prevents previous approaches from constructing more precise and discriminative item-item similarity graph. Moreover, conventional approaches always neglect group-wise item information (e.g., a item cluster containing items with similar attributes), which is critical for enhancing efficiency on message passing. Although latest multi-modal recommendation model **LGMRec** [15] employed hypergraph to exploit group-wise item information, the adopted Gumbel-Softmax mechanism is sensitive to the temperature and may fail to effectively cluster similar items.[20, 23, 47]. Hence it will inevitably incur coarse and meaningless hyperedges among group-wise item hypergraphs, leading to the limited model potentials. Therefore traditional multi-modal recommendation models cannot solve **CH1** well. On the other hand, previous cross-domain recommendation models [21, 33] fully rely on the overlapped users for knowledge sharing across domains. These approaches may not achieve satisfactory results especially when only relatively few users are overlapped [8, 35, 51]. Meanwhile some latest models (e.g., **MOTKD** [63]) could involve optimal transport techniques with Wasserstein distance [9, 50] for domain adaptation. Nonetheless, these approaches fail to consider the useful prior knowledge in domain adaptation, leading to coarse

Figure 1: The problem definition of MMCDR.

and inaccurate matching solutions [16, 37]. Therefore these strategies could result in negative transfer phenomenon [61, 70] and thus they cannot solve **CH2** well. In conclusion, previous recommendation models cannot provide satisfactory results on tackling MMCDR problem.

To address the aforementioned issues, in this paper, we propose Joint Similarity Item Exploration and Overlapped User Guidance model (**SIEOUG**), a multi-modal cross-domain recommendation model for solving MMCDR problem. **SIEOUG** includes three main modules, i.e., *similarity item exploration module, user-item collaborative filtering module* and *overlapped user guidance module*. In the similarity item exploration module, we aim to leverage pair-wise item similarity graph with proposed robust item similarity graph fusion to reduce inherit noise and achieve more reliable results. Moreover, we propose sparse item similarity hypergraph exploration in the item similarity exploration module to construct a group-wise item similarity hypergraph which enhances the efficiency of messaging passing by clustering items based on their characteristics via Gromov-Wasserstein metric. The user-item collaborative filtering module is designed to integrate graph and hypergraph information to address **CH1** and achieve user/item embeddings within each domain. In the overlapped user guidance module, we propose guidance-based optimal user matching algorithm to identify similar user pairs from a global perspective for solving **CH2**. By incorporating these three modules, **SIEOUG** can better exploit and utilize multi-modal item information, meanwhile **SIEOUG** can also fully transfer knowledge across domains sufficiently. We summarize our main contributions as follows: (1) We propose a novel framework, i.e., **SIEOUG**, for solving MMCDR problem, which contains similarity item exploration module, user-item collaborative filtering module, and overlapped user guidance module. (2) To our best knowledge, this is the first attempt in the literature to jointly explore pair-wise and group-wise item similarity via proposed robust item similarity graph aggregation and sparse item similarity hypergraph exploration. Moreover, it the first attempt to involve overlapped user guidance into knowledge transfer across domains. (3) Extensive empirical experiments on multiple tasks in Amazon datasets demonstrate the proposed **SIEOUG** significantly improves the state-of-the-art models under the MMCDR setting.

## 2 RELATED WORK

**Multi-Modal Recommendation.** Multi-Modal recommendation is set to involve multi-modal information (e.g., the item image and text descriptions) for user-item modelling [34, 40, 49, 73, 77]. Conventional work [17, 56, 69] directly fused the visual/style contents with their ID embeddings with Bayesian personalized ranking [46] for collaborative filtering. Nowadays, with the fast development of graph neural network [18], more researchers started to utilize the message passing mechanism among knowledge aggregation across different modalities [53, 55, 56]. For instance, **LATTICE** [68] first constructs an pair-wise item-item relation graph within each modality and then aggregates them to form a global item-item graph. Recently some work [75] also adopt contrastive learning strategy [52] with attention mechanism to further boost the model performance. Latest work **LGM3Rec** [15] has even adopted Gumbel-Softmax with hypergraph relationships to enhance the model's performance by capturing group-wise high-order semantic information. However, current multi-modal recommendation models fail to extract or cluster item with similar characteristics in the group-wise perspectives, which is vital for user-item modelling.

**Cross-Domain Recommendation.** Cross Domain Recommendation (CDR) models are set to tackle the data sparsity problem by leveraging useful knowledge across domains [5, 11, 27, 30, 31, 72]. Traditional CDR models mainly rely on the overlapped users to realize the knowledge transfer [6, 29, 62, 67]. For instance, **CoNet** [21] and **ACDN** [33] adopted the cross-connection unit with attention mechanism in the deep neural network for snitching the message from different domains. However these approaches cannot handle the general case when few users are overlapped [51, 65, 76]. Recently some work [66, 71] start to investigate that scenario with domain adaptation strategies including adversarial training [13] or distribution co-clustering [24]. Meanwhile, only a few papers focus on addressing the multi-modal cross-domain recommendation problem, as multi-modal information amplifies domain discrepancies, making the task even more challenging. Latest, **MOTKD** [63] is the first to attempt solving the MMCDR problem using an optimal transport approach [9, 10, 14, 26, 32]. Nonetheless, the above methods typically separate domain adaptation for overlapped and non-overlapped users, neglecting the matching guidance provided by the overlapped users. Therefore, they may lead to negative transfer, resulting in suboptimal solutions that hinder model performance.

## 3 MODELING FOR SIEOUG

In this section, we will introduce the details of our proposed method **SIEOUG** for solving MMCDR problem. We assume there are two domains, i.e., a source domain $\mathcal{S}$ and a target domain $\mathcal{T}$. There exists $N_U^{\mathcal{S}}$, $N_U^{\mathcal{T}}$ users and $N_V^{\mathcal{S}}$, $N_V^{\mathcal{T}}$ items in source and target domains respectively. $R^{\mathcal{S}} \in \mathbb{R}^{N_U^{\mathcal{S}} \times N_V^{\mathcal{S}}}$ and $R^{\mathcal{T}} \in \mathbb{R}^{N_U^{\mathcal{T}} \times N_V^{\mathcal{T}}}$ denote the user-item rating matrix in source and target domains respectively. Source and target domain inherits the origin user-item interaction graph $O^{\mathcal{S}}$ and $O^{\mathcal{T}}$ which are given as $O^{\mathcal{S}} = \begin{bmatrix} 0 & R^{\mathcal{S}} \\ (R^{\mathcal{S}})^{\top} & 0 \end{bmatrix}$ and $O^{\mathcal{T}} = \begin{bmatrix} 0 & R^{\mathcal{T}} \\ (R^{\mathcal{T}})^{\top} & 0 \end{bmatrix}$ respectively. Meanwhile items in different domains have multi-modal visual and text features. We adopt the CLIP model [45] to generate corresponding multi-modal visual and text features as $x^{\mathcal{S},\text{visual}}$, $x^{\mathcal{T},\text{visual}}$ and $x^{\mathcal{S},\text{text}}$, $x^{\mathcal{T},\text{text}}$ respectively. Not all users are overlapped across domains and we adopt $\mathcal{K}_u \in (0, 1)$ as the

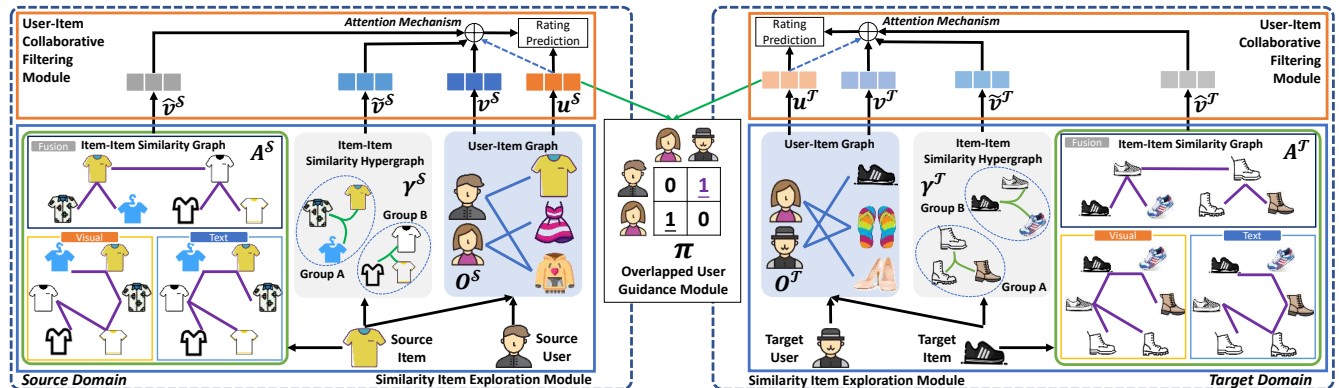

Figure 2: The model framework of proposed SIEOUG for solving MMCDR problem.

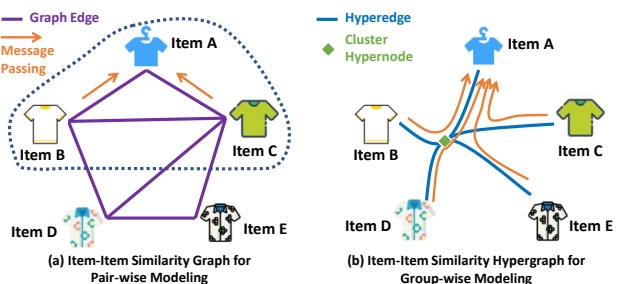

Figure 3: The illustrations on message passing on item-item similarity graph and hypergraph. Obviously, hypergraph can aggregate more useful information during the procedure.

overlapped user ratio to measure how many users are concurrence among different domains following [24].

The framework of **SIEOUG** is shown in Fig.2. **SIEOUG** mainly has three modules, i.e., *similarity item exploration module, user-item collaborative filtering module* and *overlapped user guidance module*. Similarity item exploration module is set to exploit multi-modal information with vision and text features for different items within each domain. User-item collaborative filtering module aims to model user and item preferences based on user-item ratings. Overlapped user guidance module is set to transfer and aggregate useful knowledge across different domains. Combining similarity item exploration module and overlapped user guidance module, **SIEOUG** can better explore both intra-domain multi-modal information and inter-domain user sharing information.

## 3.1 Similarity Item Exploration Module

To start with, we first introduce similarity item exploration module in **SIEOUG**. Similarity item exploration module includes two main steps, i.e., *pair-wise item similarity graph construction* and *group-wise item similarity hypergraph construction*. Pair-wise item similarity graph construction aims to denoise pair-wise item relationship among diverse multi-modal information. Group-wise item similarity hypergraph construction aims to cluster items based on their characteristics to enhance the efficiency on message passing. By combining item similarity graph and hypergraph construction, the intra-domain model can fully explore item characteristics for enhancing the performance.

**Pair-Wise Item Similarity Graph Construction.** Following previous multi-modal work [68, 74], we should calculate the item pair-wise similarity $S_{ij}^{d,m}$ among different modality in the first step of robust item similarity graph construction, i.e., for each modality $m$:

$$S_{ij}^{d,m} = \exp\left[\frac{(x_i^{d,m})^\top x_j^{d,m}}{\|x_i^{d,m}\|_2 \|x_j^{d,m}\|_2}\right] = \mathcal{S}(x_i^{d,m}, x_j^{d,m}), \quad (1)$$

where $d = \{\mathcal{S}, \mathcal{T}\}$ indicates the domain index and $m = \{\text{visual, text}\}$ denotes the visual and text features. $x^{d,m}$ denotes the $m$ modality of features in domain $d$ and $\mathcal{S}(\cdot)$ denotes the similarity calculation. Then we adopt top-$z$ sparsification method to obtain sparse similarity results via filtering out irrelevant item-item relationships:

$$\widehat{S}_{ij}^{d,m} = \begin{cases} 1, & S_{ij}^{d,m} \in \text{top}_z\left(S_{i*}^{d,m}\right) \\ 0, & \text{otherwise.} \end{cases} \quad (2)$$

We can obtain different sparse similarity results $\widehat{S}^{d,m}$ according to different modalities in domain $d$. Since different modalities may inherit complex and heterogeneous information [74], we should aggregate them to figure out general pair-wise item similarity $A^d \in \mathbb{R}^{N \times N}$. To fulfill this task, we propose Robust Item Similarity Graph Fusion (RISGF) method to find $A^d$ which can be shown as:

$$\min_{A^d, \Delta^{d,m}} Q^d = \frac{1}{M} \sum_{m=1}^{M} \left(\frac{1}{2}\left\|\left(A^d + \Delta^{d,m}\right) - \widehat{S}^{d,m}\right\|_2^2 + \mu\left\|\Delta^{d,m}\right\|_1\right)$$

$$s.t. \ A_{ij}^d \in [0, 1], \quad (3)$$

where $\Delta^{d,m} \in \mathbb{R}^{N \times N}$ denotes modality-specific information in each domain and $\mu$ denotes the balanced hyper parameter. Ideally, pair-wise item similarity within each modality $\widehat{S}^{d,m}$ can be viewed as the combination of general pair-wise item similarity $A^d$ with modality-specific information $\Delta^{d,m}$. Thus RISGF can exploit more relevant pair-wise item similarity while filtering out irrelevant bias or noise in $\widehat{S}^{d,m}$. Inspired by previous works [3, 58], we adopt L1-norm on $\widehat{S}^{d,m}$ to make the result of $A^d$ more robust.

**Optimization.** We provide the optimization details on RISGF as below. Firstly, we fix $A^d$ and update modality-specific information

$\Delta^{d,m}$ via proximal gradient descent [43] accordingly:

$$\Delta_{ij}^{d,m} = \mathcal{T}_\mu(\widehat{S}_{ij}^{d,m} - A_{ij}^d) = \begin{cases} \widehat{S}_{ij}^{d,m} - A_{ij}^d - \mu, & \widehat{S}_{ij}^{d,m} \geq A_{ij}^d + \mu \\ \widehat{S}_{ij}^{d,m} - A_{ij}^d + \mu, & \widehat{S}_{ij}^{d,m} \leq A_{ij}^d - \mu \\ 0, & \text{otherwise,} \end{cases} \quad (4)$$

where $\mathcal{T}_\mu(\cdot)$ denotes the proximal projection operator. Then we can fix $\Delta^{d,m}$ and update $A^d$ as below:

$$A^d = \left[ \frac{1}{M} \sum_{m=1}^{M} \left( \widehat{S}_{ij}^{d,m} - \Delta^{d,m} \right) \right]_*, \quad (5)$$

where $[y]_* = \max(\min(y, 1), 0)$ denotes the projected operation. We can adopt Eq.(4) and Eq.(5) iteratively to solve RISGA for obtaining the optimal solution on $A^d$ and $\Delta^{d,m}$. At that time, the general pair-wise item similarity $A^d$ can effectively incorporate multi-modal information with enhanced robustness for modelling.

**Group-Wise Item Similarity Hypergraph Construction.** Although pair-wise item similarity graph construction has provided item similarity matrix $A^d$, it still mainly relies on pair-wise information with low efficiency for messaging passing [64]. Therefore, it is essential to further explore group-wise item similarity relationship. That is, we should figure out the items with similar characteristics to construct the corresponding hypergraph $\gamma^d$ via the group-wise item clustering. Hence we propose Sparsity Item Similarity Hypergraph Exploration (SISHE) method to find $\gamma^d$ accordingly:

$$\min_{\gamma^d} Q^d = - \left\langle A^d \gamma^d I_K, \gamma^d \right\rangle + \frac{\eta}{2} ||\gamma^d||_2^2$$

$$s.t. \sum_{i=1}^{N} \gamma_{ij}^d = \frac{N}{K}, \quad \sum_{j=1}^{K} \gamma_{ij}^d = 1, \quad \gamma_{ij}^d \geq 0, \quad (6)$$

where $\eta$ denotes the hyper parameter. $I_K \in \mathbb{R}^{K \times K}$ denotes the identity matrix and $K$ denotes the number of clusters. $\gamma^d \in \mathbb{R}^{N \times K}$ is the clustering matrix for constructing group-wise item similarity hypergraph. SISHE aims to balance cluster items based on the pair-wise item similarity graph $A^d$ via optimizing Eq.(6).

**Optimization.** To start with, we first initialize $(\gamma^d)_{ij}^{(0)} = \frac{1}{K}$. Then we regard $(-A^d(\gamma^d)^{(l)} I_K) = \mathcal{Y}^{(l)}$ as the constant during the $l$-th iteration. We can provide the Lagrange multipliers of Eq.(6) at the $l$-th iteration as follows:

$$\min_{(\gamma^d)^{(l+1)}} (J^d)^{(l+1)} = \left\langle \mathcal{Y}^{(l)}, (\gamma^d)^{(l+1)} \right\rangle + \frac{\eta}{2} ||(\gamma^d)^{(l+1)}||_2^2$$

$$- \sum_{j=1}^{K} g_j \left( \sum_{i=1}^{N} \gamma_{ij}^d - \frac{N}{K} \right) - \sum_{i=1}^{N} f_i \left( \sum_{j=1}^{K} \gamma_{ij}^d - 1 \right), \quad (7)$$

where $f$ and $g$ denote the Lagrange multipliers. By taking the differentiation w.r.t. on $(\gamma^d)_{ij}^{(l+1)}$ and set it as 0, we can obtain the following equations:

$$(\gamma^d)_{ij}^{(l+1)} = \left[ \frac{f_i + g_j - \mathcal{Y}_{ij}^{(l)}}{\eta} \right]_+, \quad (8)$$

where $[e]_+ = \max(0, e)$. Specifically, we first fix $g$ and update $f$ via solving the equation $\phi(f_i) = \sum_{j=1}^{K} [f_i - (\mathcal{Y}_{ij}^{(l)} - g_j)]_+ = \eta$. That is, we sort $(\mathcal{Y}_{ij}^{(l)} - g_j)$ in ascending order and define it as the reordered vector $\mathcal{G}^{(l)}$, i.e, $\mathcal{G}_o^{(l)} \leq \mathcal{G}_{o+1}^{(l)} \forall_o$. Therefore, the result of $f$ can be

shown as:

$$f_i = \begin{cases} \mathcal{G}_o^{(l)} + \frac{\eta - \phi(\mathcal{G}_o^{(l)})}{\phi(\mathcal{G}_{o+1}^{(l)}) - \phi(\mathcal{G}_o^{(l)})} (\mathcal{G}_{o+1}^{(l)} - \mathcal{G}_o^{(l)}), & \phi(\mathcal{G}_o^{(l)}) \leq \eta < \phi(\mathcal{G}_{o+1}^{(l)}) \\ \frac{1}{K} \left( \eta + \sum_{j=1}^{K} \mathcal{G}_j^{(l)} \right), & \eta \geq \phi(\mathcal{G}_K^{(l)}). \end{cases}$$

$$(9)$$

Likewise, we then fix $f$ and update $g$ via solving the equation $\psi(g_j) = \sum_{i=1}^{N} [g_j - (\mathcal{Y}_{ij}^{(l)} - f_i)]_+ = \eta \frac{N}{K}$. That is, we sort $(\mathcal{Y}_{ij}^{(l)} - f_i)$ in ascending order and define the reordered vector as $\mathcal{F}^{(l)}$. Hence the result of $g$ can be shown as:

$$g_j = \begin{cases} \mathcal{F}_o^{(l)} + \frac{\eta \frac{N}{K} - \psi(\mathcal{F}_o^{(l)})}{\psi(\mathcal{F}_{o+1}^{(l)}) - \psi(\mathcal{F}_o^{(l)})} (\mathcal{F}_{o+1}^{(l)} - \mathcal{F}_o^{(l)}), & \psi(\mathcal{F}_o^{(l)}) \leq \eta \frac{N}{K} < \psi(\mathcal{F}_{o+1}^{(l)}) \\ \frac{1}{N} \left( \eta \frac{N}{K} + \sum_{i=1}^{N} \mathcal{F}_j^{(l)} \right), & \eta \frac{N}{K} \geq \psi(\mathcal{F}_K^{(l)}). \end{cases}$$

$$(10)$$

After several inner iterations via Eq.(9) and Eq.(10), we can obtain the optimal solutions on $f$ and $g$ for $(\gamma^d)^{(l+1)}$. Then we should update the term $(-A^d(\gamma^d)^{(l+1)} I_K) = \mathcal{Y}^{(l+1)}$ for the outer iteration to solve $(\gamma^d)^{(l+2)}$. Finally we can reach the optimal coupling matrix $\gamma^d$ in each domain $d$. The pseudo algorithm for SISHE is provided in Appendix A. Indeed, $\gamma^d$ denotes the clustering results of items and thus $\gamma^d$ can be regarded as the item-item hypergraph and each hyperedge represents the items with similar characteristics. Moreover, SISHE with Gromov-Wasserstein metric [39, 44, 60] can avoid coarse and inaccurate relationships comparing when compared to prior methods [12, 59] that mainly rely on the Gumbel-Softmax mechanism [20, 47, 57]. In summary, the group-wise item hypergraph can aggregate more information compared to the pair-wise item similarity graph through the message passing procedure, as illustrated in Fig.3(a)-(b). Therefore adopting group-wise item hypergraph can further enhance the model performance.

## 3.2 User-Item Collaborative Filtering Module

After we obtain user-item interaction graph $O^d$, item-item similarity graph $A^d$ and item-item clustering hypergraph $\gamma^d$ in each domain, we should combine these useful information for user-item modelling. Therefore, we propose user-item collaborative filtering module to fully exploit user and item embeddings for the recommendation. Specifically, we set the user/item initialize ID embeddings as $u_d^{(0)}$ and $v_d^{(0)}$ via the learnable lookup table in domain $d$ with dimension $\mathcal{D}$ respectively. Firstly, we adopt the graph neural network on user-item interaction graph $O^d$ at the $l$-th layer as follows:

$$e_d^{(l+1)} = (D_O^d)^{-\frac{1}{2}} O^d (D_O^d)^{-\frac{1}{2}} e_d^{(l)}, \quad (11)$$

where $e_d^{(l)} = u_d^{(l)} \| v_d^{(l)}$ and $\|$ denotes the concatenate operation. $D_O^d = \text{diag}(O^d \mathbf{1})$ denotes degree matrix of $O^d$. The user/item rating preference embeddings after $\ell$-layered graph neural network is given as $u^d = \frac{1}{\ell+1} \sum_{l=0}^{\ell} u_d^{(\ell)}$ and $v^d = \frac{1}{\ell+1} \sum_{l=0}^{\ell} v_d^{(\ell)}$ respectively. Meanwhile we should further utilize multi-modal information among intra-domain items. That is we conduct the graph convolution according to item-item similarity graph $A^d$ as:

$$\widehat{v}_d^{(l+1)} = (D_S^d)^{-\frac{1}{2}} A^d (D_S^d)^{-\frac{1}{2}} \widehat{v}_d^{(l)}, \quad (12)$$

where $\widehat{v}_d^{(0)} = v_d^{(0)}$ and $D_S^d = \text{diag}(A^d \mathbf{1})$ denotes degree matrix of $A^d$. Likewise, we can obtain the item similarity-based embedding

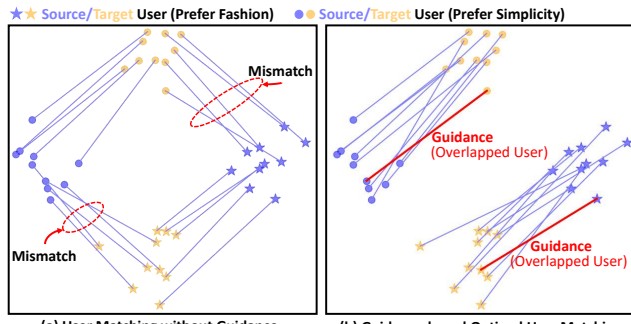

**Figure 4: Illustrations on overlapped user guidance. When we involve prior overlapped users as the guidance, we can avoid mismatches between users with different preferences.**

via $\widetilde{\boldsymbol{v}}^d = \frac{1}{\ell+1} \sum_{l=0}^{\ell} \widetilde{\boldsymbol{v}}_d^{(\ell)}$. Moreover, we conduct the hypergraph based on $\boldsymbol{\gamma}^d$ to aggregate group-wise information as follows:

$$
\begin{aligned}
\widetilde{\boldsymbol{v}}_d^{(l+1)} &= \text{HyperGraph}(\boldsymbol{\gamma}^d, \widetilde{\boldsymbol{v}}_d^{(l)}, \boldsymbol{W}_d^{(l)}) \\
&= (\boldsymbol{D}_V^d)^{-\frac{1}{2}} (\boldsymbol{\gamma}^d) (\boldsymbol{D}_E^d)^{-\frac{1}{2}} (\boldsymbol{\gamma}^d)^{\top} (\boldsymbol{D}_V^d)^{-\frac{1}{2}} \widetilde{\boldsymbol{v}}_d^{(l)} \boldsymbol{W}_d^{(l)},
\end{aligned}
\tag{13}
$$

where $\widetilde{\boldsymbol{v}}_d^{(0)} = \boldsymbol{v}_d^{(0)}$ and $\boldsymbol{D}_V^d \in \mathbb{R}^{N \times N}$ and $\boldsymbol{D}_E^d \in \mathbb{R}^{K \times K}$ are vertex degrees and hyperedges degrees of items to cluster hypergraphs. $\boldsymbol{W}_d^{(l)} \in \mathbb{R}^{N \times \mathcal{D}}$ denotes the learnable weights at the $l$-th layer in hypergraph. We can obtain the item cluster-based embedding via $\widetilde{\boldsymbol{v}}^d = \frac{1}{\ell+1} \sum_{l=0}^{\ell} \widetilde{\boldsymbol{v}}_d^{(\ell)}$. Therefore we can achieve user and item embedding $\boldsymbol{\mathcal{U}}^d$ and $\boldsymbol{\mathcal{V}}^d$ by joining multi-modal information as:

$$
\boldsymbol{\mathcal{U}}^d = \boldsymbol{u}^d, \quad \boldsymbol{\mathcal{V}}^d = \alpha^d \boldsymbol{v}^d + \widehat{\alpha}^d \widetilde{\boldsymbol{v}}^d + \widetilde{\alpha}^d \widetilde{\boldsymbol{v}}^d, \tag{14}
$$

where $\alpha^d$, $\widehat{\alpha}^d$ and $\widetilde{\alpha}^d$ denotes the normalized adaptive weights as $\alpha^d + \widehat{\alpha}^d + \widetilde{\alpha}^d = 1$. Using $\alpha^d$ as an example, it can be calculated as:

$$
\alpha^d = \frac{\exp(\boldsymbol{W}_u^d \boldsymbol{u}^d + \boldsymbol{W}_v^d \boldsymbol{v}^d)}{\sum_{\chi \in \{\boldsymbol{v}, \widetilde{\boldsymbol{v}}, \widehat{\boldsymbol{v}}\}} \exp(\boldsymbol{W}_u^d \boldsymbol{u}^d + \boldsymbol{W}_\chi^d \boldsymbol{\chi}^d)}, \tag{15}
$$

where $\boldsymbol{W}_u^d$, $\boldsymbol{W}_v^d$, $\boldsymbol{W}_{\widehat{v}}^d$ and $\boldsymbol{W}_{\widetilde{v}}^d$ denote the learnable weights. After that, we can adopt the commonly-used contrastive collaborative filtering loss for modelling intra-domain user-item ratings:

$$
\min L_R^d = -\log(\mathcal{S}(\boldsymbol{\mathcal{U}}_i^d, \boldsymbol{\mathcal{V}}_j^d)) + \log \left( \sum_{\boldsymbol{\mathcal{V}}_k^d \in \mathcal{N}(\boldsymbol{\mathcal{U}}_i^d)} \mathcal{S}(\boldsymbol{\mathcal{U}}_i^d, \boldsymbol{\mathcal{V}}_k^d) \right). \tag{16}
$$

That is, we aim to can pull the positive items close and push the negative items away for a certain user according to his/her preference in each domain $d$ via the loss function $L_R^d$.

## 3.3 Overlapped User Guidance Module

Although combining similarity item exploration module and user-item collaborative filtering module can enhance intra-domain model performance, it still cannot share and transfer useful knowledge across different domains with limited overlapped users. Hence we further propose overlapped user guidance module to utilize inter-domain information for tackling data sparsity problem in each domain. Previous CDR methods [21, 33] mainly focus on the knowledge sharing via limited overlapped users and neglecting the majority part of non-overlapped users. Thus these conventional approaches cannot better resolve the issue of knowledge transfer well.

Therefore, **SIEOUG** should leverage knowledge transfer among all users by utilizing overlapping users as the useful guidance from a global optimal perspective. Specifically, we first propose Guidance-based Optimal User Matching (GOUM) method among the whole source and target users during the training procedure:

$$
\min_{\boldsymbol{\pi} \geq 0} J_{\text{agn}} = \langle \boldsymbol{\pi}, \boldsymbol{C} \odot \boldsymbol{M} \rangle = \langle \boldsymbol{\pi}, \boldsymbol{Q} \rangle \quad s.t. \sum_{j=1}^{N} \pi_{ij} = 1, \sum_{i=1}^{N} \pi_{ij} = 1, \tag{17}
$$

where $C_{ij} = ||\boldsymbol{\mathcal{U}}_i^S - \boldsymbol{\mathcal{U}}_j^{\mathcal{T}}||_2^2$ denotes the embedding distance between the source and target users and $\boldsymbol{\pi} \in \mathbb{R}^{N \times N}$ denotes the user-user matching matrix. $\boldsymbol{M} \in \mathbb{R}^{N \times N}$ denotes the masked matrix which can be calculated as $\mathcal{M}_{ij} = 1 - \delta(u_i^S, u_j^{\mathcal{T}})$ where $\delta(u_i^S, u_j^{\mathcal{T}})$ denotes whether $u_i^S$ and $u_j^{\mathcal{T}}$ are overlapped users. Specifically, if $u_i^S$ and $u_j^{\mathcal{T}}$ are overlapped users, $\delta(u_i^S, u_j^{\mathcal{T}}) = 1$ and otherwise $\delta(u_i^S, u_j^{\mathcal{T}}) = 0$. $\boldsymbol{M}$ can guarantee that the overlapped users should get matched which conforms to our anticipation and it can further provides useful prior information as the guidance signal in solving GOUM. As we show an example in Fig.4(a) where we will obtain mismatch user matching with different preferences when we do not consider any prior knowledge (e.g., overlapped users) as guidance. When we consider the overlapped user as guidance via the masked matrix during the matching procedure with the red line segments, we can achieve clear results as shown in Fig.4(b).

**Optimization.** To start with, we first figure out the Lagrange multipliers of GOUM as below:

$$
\max_{\boldsymbol{\omega}, \boldsymbol{\kappa}, s} \min_{\boldsymbol{\pi}} \widehat{J}_{\text{agn}} = \langle \boldsymbol{\pi}, \boldsymbol{Q} \rangle - \langle \boldsymbol{\omega}, \boldsymbol{\pi} \mathbf{1}_N - \mathbf{1}_N \rangle - \langle \boldsymbol{\kappa}, \boldsymbol{\pi}^\top \mathbf{1}_N - \mathbf{1}_N \rangle - \langle \boldsymbol{\pi}, \boldsymbol{s} \rangle, \tag{18}
$$

where $\boldsymbol{\omega}$, $\boldsymbol{\kappa}$ and $\boldsymbol{s}$ denote the Lagrange multipliers. Meanwhile $\boldsymbol{s}$ satisfies $s_{ij} \geq 0$ and $s_{ij} \pi_{ij} = 0$ according to the KKT conditions. Therefore, it is obvious to achieve the dual form of GOUM:

$$
\max_{\boldsymbol{\omega}, \boldsymbol{\kappa}} \mathcal{J} = \langle \boldsymbol{\omega}, \boldsymbol{a} \rangle + \langle \boldsymbol{\kappa}, \boldsymbol{b} \rangle \quad s.t. \omega_i + \kappa_j + s_{ij} = Q_{ij}, s_{ij} \geq 0, \tag{19}
$$

where we can observe that $\omega_i + \kappa_j \leq Q_{ij}$. That is, we can adopt $c$-transform to obtain $\kappa_j = \inf_{k \in [N]}(Q_{kj} - \omega_k)$. Although we should only need to optimize variable $\boldsymbol{\omega}$ in Eq.(19), the optimization problem is non-smooth. Hence we should adopt the smoothness approximation [41] on $\boldsymbol{\kappa}$ to facilitate the following procedure:

$$
\inf_{k \in [N]} \left( Q_{kj} - \omega_k \right) = -\lim_{\epsilon \to 0} \left[ \epsilon \log \left[ \sum_{i=1}^{N} \exp \left( \frac{\omega_i - Q_{ij}}{\epsilon} \right) \right] \right]. \tag{20}
$$

At that time, the optimization in Eq.(19) can be rewritten with the following approximate but smoothness expression:

$$
\begin{aligned}
\min_{\widehat{\boldsymbol{\omega}}} \mathcal{J}_{\text{KP}} &= \sum_{j=1}^{N} \left\{ \epsilon \log \left[ \sum_{i=1}^{N} \exp \left( \frac{\widehat{\omega}_i - Q_{ij}}{\epsilon} \right) \right] \right\} - \sum_{i=1}^{N} \widehat{\omega}_i \\
&= -\sum_{j=1}^{N} \widehat{\kappa}_j - \sum_{i=1}^{N} \widehat{\omega}_i,
\end{aligned}
\tag{21}
$$

where $\widehat{\boldsymbol{\kappa}}$ and $\widehat{\boldsymbol{\omega}}$ denotes the approximation on $\boldsymbol{\kappa}$ and $\boldsymbol{\omega}$ respectively. When $\epsilon \to 0$, $\widehat{\boldsymbol{\kappa}}$ and $\widehat{\boldsymbol{\omega}}$ will be relatively close to the true value of $\boldsymbol{\kappa}$ and $\boldsymbol{\omega}$ [1]. We can adopt FISTA algorithm [2] to optimize Eq.(21) efficiently for achieving the optimal solution on $\widehat{\boldsymbol{\kappa}}$, $\widehat{\boldsymbol{\omega}}$ and $\widehat{s}_{ij} = Q_{ij} - \widehat{\omega}_i - \widehat{\kappa}_j$. Note that ideally when $\widehat{s}_{ij} = 0$, it indicates that $\pi_{ij} = 1$ according to the KKT theorem. In practice, we select the

smallest value of $\widehat{s}_{i*}$ for the $i$-th row as the actual matching results:

$$\pi_{ij} = \begin{cases} 1, & \arg\min_j \left[Q_{ij} - \widehat{\omega}_i - \widehat{\kappa}_j\right], \\ 0, & \text{otherwise.} \end{cases} \quad (22)$$

The pseudo algorithm for GOUM is provided in Appendix B. After we obtain the user-user matching matrix $\boldsymbol{\pi}^*$, we can determine the most relevant users across domains among the whole set of users. That is for the source user $\boldsymbol{\mathcal{U}}^{\mathcal{S}}$, the most relevant target users is given as $\boldsymbol{\pi}\boldsymbol{\mathcal{U}}^{\mathcal{T}}$. Likewise, for the target user $\boldsymbol{\mathcal{U}}^{\mathcal{T}}$, the most relevant source users is given as $\boldsymbol{\pi}^{\top}\boldsymbol{\mathcal{U}}^{\mathcal{S}}$. Therefore, we further propose guidance user contrastive loss across domains:

$$\min L_C = -\sum_{i=1}^{N}\left[\log(\mathcal{S}(\boldsymbol{\mathcal{U}}_i^{\mathcal{S}}, [\boldsymbol{\pi}\boldsymbol{\mathcal{U}}^{\mathcal{T}}]_i)) + \log(\sum_{\boldsymbol{\mathcal{U}}_k^{\mathcal{T}} \in \mathcal{H}(\boldsymbol{\mathcal{U}}_i^{\mathcal{S}})}\mathcal{S}(\boldsymbol{\mathcal{U}}_i^{\mathcal{S}}, \boldsymbol{\mathcal{U}}_k^{\mathcal{T}}))\right]$$

$$-\sum_{j=1}^{N}\left[\log(\mathcal{S}(\boldsymbol{\mathcal{U}}_j^{\mathcal{T}}, [\boldsymbol{\pi}^{\top}\boldsymbol{\mathcal{U}}^{\mathcal{S}}]_j)) + \log(\sum_{\boldsymbol{\mathcal{U}}_k^{\mathcal{S}} \in \mathcal{H}(\boldsymbol{\mathcal{U}}_j^{\mathcal{T}})}\mathcal{S}(\boldsymbol{\mathcal{U}}_j^{\mathcal{T}}, \boldsymbol{\mathcal{U}}_k^{\mathcal{S}}))\right],$$

$$(23)$$

where $\mathcal{H}(\boldsymbol{\mathcal{U}}_i^{\mathcal{S}}) = \boldsymbol{\mathcal{U}}^{\mathcal{T}} \setminus [\boldsymbol{\pi}\boldsymbol{\mathcal{U}}^{\mathcal{T}}]_i$ and $\mathcal{H}(\boldsymbol{\mathcal{U}}_j^{\mathcal{T}}) = \boldsymbol{\mathcal{U}}^{\mathcal{S}} \setminus [\boldsymbol{\pi}^{\top}\boldsymbol{\mathcal{U}}^{\mathcal{S}}]_j$ denote irrelevant users which act as the negative samples. The guidance contrastive loss can enforce users with similar characteristics across domains to move closer in the embedding space. Thus the inter-domain knowledge can enhance the model training.

### 3.4 Putting Together

The total loos of **SIEOUG** can be obtained by the contrastive collaborative filtering loss and guidance user contrastive loss on source and target domains. That is, the loss of **SIEOUG** is given as:

$$\min L = L_R^{\mathcal{S}} + L_R^{\mathcal{T}} + \lambda L_C, \quad (24)$$

where $\lambda$ denotes the balanced hyper parameters. By doing this, **SIEOUG** can not only model items with diverse modalities, but also aggregate useful knowledge among the guidance of overlapped users across domains. Note that we pre-calculate the pair-wise item-item similarity graph $A^d$ via RISGF beforehand and freeze them during the training stage to reduce the computation burden.

## 4 EMPIRICAL STUDY

In this section, we conduct experiments on several real-world datasets to answer the following questions: (1) **RQ1**: How does the proposed model **SIEOUG** perform compared with the state-of-the-art recommendation methods? (2) **RQ2**: How do the similarity item exploration and overlapped user guidance contribute to performance improvement? (3) **RQ3**: Can proposed method be extended to collaborate with other recommendation models? (4) **RQ4**: How does the performance of **SIEOUG** vary with different values of the hyper-parameters?

### 4.1 Datasets and Tasks

We conduct extensive experiments on commonly used real-world **Amazon** datasets. The **Amazon** dataset [42] has five domains, i.e., Clothing Shoes and Jewelry (Clothes), Sports and Outdoors (Sports), Tools and Home Improvement (Home), Grocery and Gourmet Food (Food), and Home and Kitchen (Kitchen). Each datasets includes user-item rating interactions, item text descriptions and item image

information. Specifically, we conduct three different tasks on MM-CDR, e.g., (Task 1) Clothes ↔ Sports, (Task 2) Clothes ↔ Home, (Task 3) Food ↔ Kitchen. For each dataset, we binarize the ratings higher or equal to 4 as 1 and the rest as 0 following [24]. Then we filter the users and items with less than 10 interactions. The detailed datasets statistics after prepossessing are shown in Appendix.C.

### 4.2 Experiment Settings

We randomly divide the user-item rating data into training, validation, and test sets with a ratio of 8:1:1 following [74]. We also adjust the overlapped user ratio $\mathcal{K}_u$ in $\{10\%, 50\%, 90\%\}$ to verify the effectiveness our model following [24, 25]. When $\mathcal{K}_u$ is larger (e.g., $\mathcal{K}_u = 90\%$), the problem is much easier due to the ample guidance provided by the overlapping users. In contrast, when $\mathcal{K}_u$ is smaller (e.g., $\mathcal{K}_u = 10\%$), the problem becomes significantly more challenging. We set batch size $N = 256$ for training in both source and target domains. We set $z = 5$ for the top-$z$ sparsification in the pair-wise item similarity graph construction. We set $\mu = 0.1$ and $\eta = 0.1$ for RISGF and SISHE respectively. Meanwhile we set the number of item clusters as $K = 15$ for constructing the item hypergraph. The number of layers for both graph and hypergraph convolution networks are set to 3. The latent dimension of user and item embeddings are given as $D = 128$. We set $\epsilon = 0.01$ for GOUM in the overlapped user guidance module. Finally we set $\lambda = 0.6$ for the loss function in **SIEOUG**. For all the experiments, we perform five random experiments and report the average results. We choose Adam [28] as optimizer, and adopt HR@$k$ and NDCG@$k$ [54] as the ranking evaluation metrics with $k = 10$.

**Baseline.** We compare our proposed **SIEOUG** with the following state-of-the-art models. (1) **NeuMF** [19] first utilizes deep neural networks for collaborative filtering in the single domain. (2) **LightGCN** [18] first adopts graph convolution network to further enhance intra-domain model performance. (3) **LATTICE** [68] first exploits the similarity graph among multi-modal item information for user-item modeling. (4) **FREEDOM** [74] further aggregates multi-modal information with denoising mechanism. (5) **LGM3Rec** [15] utilizes local and global graph learning methods for multimodal recommendation. (6) **DisenCDR** [4] adopts variational disentangle mechanism for cross domain recommendation among overlapped users. (7) **ETL** [7] utilizes dual autoencoder framework with equivalent transformation component across domains. (8) **VDEA** [24] exploits domain-invariant user preferences with co-clustering methods across domains. (9) **DURation** [35] adopts distribution variance and correlation alignment to obtain unified representations. (10) **CL-DTCDR** [36] adopt dual-target contrastive learning method for cross domain recommendation. (11) **MOTKD** [63] is the state-of-the-art model which adopting optimal transport for multi-modal cross domain recommendation.

### 4.3 Recommendation Performance

**Results and discussion.** The comparison results on different datasets are shown in Table.1. From that we can observe: (1) Single domain multi-modal recommendation models (e.g., **LATTICE** and **FREEDOM**) can enhance the model performance against conventional recommendation models (e.g., **LightGCN**). However, these single-domain models cannot leverage useful knowledge across

**Table 1: Experimental results on Amazon datasets.**

| Clothes & Sports | Clothes ($\mathcal{K}_u = 10\%$) | | Sports ($\mathcal{K}_u = 10\%$) | | Clothes ($\mathcal{K}_u = 50\%$) | | Sports ($\mathcal{K}_u = 50\%$) | | Clothes ($\mathcal{K}_u = 90\%$) | | Sports ($\mathcal{K}_u = 90\%$) | |
|---|---|---|---|---|---|---|---|---|---|---|---|---|
| | HR@10 | NDCG@10 | HR@10 | NDCG@10 | HR@10 | NDCG@10 | HR@10 | NDCG@10 | HR@10 | NDCG@10 | HR@10 | NDCG@10 |
| NeuMF | 0.0383 | 0.0131 | 0.0490 | 0.0275 | 0.0564 | 0.0202 | 0.0779 | 0.0265 | 0.0870 | 0.0316 | 0.1112 | 0.0488 |
| LightGCN | 0.0429 | 0.0205 | 0.0573 | 0.0316 | 0.0682 | 0.0264 | 0.0851 | 0.0396 | 0.0968 | 0.0403 | 0.1184 | 0.0573 |
| LATTICE | 0.0558 | 0.0297 | 0.0626 | 0.0364 | 0.0753 | 0.0341 | 0.0930 | 0.0471 | 0.1003 | 0.0489 | 0.1256 | 0.0668 |
| FREEDOM | 0.0636 | 0.0354 | 0.0725 | 0.0404 | 0.0779 | 0.0374 | 0.0954 | 0.0527 | 0.1031 | 0.0542 | 0.1283 | 0.0707 |
| LGM3Rec | 0.0645 | 0.0363 | 0.0739 | 0.0421 | 0.0806 | 0.0417 | 0.0992 | 0.0543 | 0.1054 | 0.0568 | 0.1307 | 0.0723 |
| DisenCDR | 0.0565 | 0.0252 | 0.0667 | 0.0341 | 0.0780 | 0.0389 | 0.0968 | 0.0526 | 0.1097 | 0.0592 | 0.1361 | 0.0740 |
| ETL | 0.0594 | 0.0286 | 0.0701 | 0.0378 | 0.0812 | 0.0405 | 0.1013 | 0.0540 | 0.1066 | 0.0564 | 0.1347 | 0.0751 |
| VDEA | 0.0653 | 0.0351 | 0.0724 | 0.0402 | 0.0846 | 0.0432 | 0.1039 | 0.0575 | 0.1102 | 0.0587 | 0.1368 | 0.0743 |
| DURation | 0.0641 | 0.0338 | 0.0742 | 0.0439 | 0.0875 | 0.0423 | 0.1057 | 0.0601 | 0.1124 | 0.0610 | 0.1395 | 0.0762 |
| CL-DTCDR | 0.0679 | 0.0368 | 0.0755 | 0.0472 | 0.0903 | 0.0441 | 0.1084 | 0.0612 | 0.1136 | 0.0621 | 0.1408 | 0.0775 |
| MOTKD | 0.0705 | 0.0375 | 0.0781 | 0.0493 | 0.0939 | 0.0452 | 0.1114 | 0.0623 | 0.1167 | 0.0640 | 0.1432 | 0.0806 |
| **SIEOUG**-O | 0.0684 | 0.0390 | 0.0769 | 0.0467 | 0.0892 | 0.0437 | 0.1061 | 0.0578 | 0.1115 | 0.0603 | 0.1364 | 0.0776 |
| **SIEOUG**-M | 0.0722 | 0.0403 | 0.0805 | 0.0517 | 0.0961 | 0.0498 | 0.1174 | 0.0650 | 0.1250 | 0.0676 | 0.1535 | 0.0884 |
| **SIEOUG**-G | 0.0740 | 0.0409 | 0.0826 | 0.0531 | 0.0972 | 0.0503 | 0.1185 | 0.0661 | 0.1254 | 0.0667 | 0.1528 | 0.0875 |
| **SIEOUG** | **0.0808** | **0.0437** | **0.0873** | **0.0595** | **0.1034** | **0.0543** | **0.1276** | **0.0716** | **0.1292** | **0.0692** | **0.1551** | **0.0899** |

| Clothes & Home | Clothes ($\mathcal{K}_u = 10\%$) | | Home ($\mathcal{K}_u = 10\%$) | | Clothes ($\mathcal{K}_u = 50\%$) | | Home ($\mathcal{K}_u = 50\%$) | | Clothes ($\mathcal{K}_u = 90\%$) | | Home ($\mathcal{K}_u = 90\%$) | |
|---|---|---|---|---|---|---|---|---|---|---|---|---|
| | HR@10 | NDCG@10 | HR@10 | NDCG@10 | HR@10 | NDCG@10 | HR@10 | NDCG@10 | HR@10 | NDCG@10 | HR@10 | NDCG@10 |
| NeuMF | 0.0498 | 0.0181 | 0.0234 | 0.0113 | 0.0755 | 0.0269 | 0.0517 | 0.0173 | 0.1084 | 0.0536 | 0.0702 | 0.0245 |
| LightGCN | 0.0635 | 0.0267 | 0.0392 | 0.0178 | 0.0843 | 0.0381 | 0.0664 | 0.0254 | 0.1196 | 0.0605 | 0.0827 | 0.0363 |
| LATTICE | 0.0747 | 0.0354 | 0.0511 | 0.0259 | 0.0970 | 0.0463 | 0.0739 | 0.0325 | 0.1304 | 0.0682 | 0.0916 | 0.0421 |
| FREEDOM | 0.0810 | 0.0404 | 0.0578 | 0.0312 | 0.1062 | 0.0557 | 0.0796 | 0.0369 | 0.1353 | 0.0739 | 0.0943 | 0.0488 |
| LGM3Rec | 0.0843 | 0.0429 | 0.0602 | 0.0336 | 0.1091 | 0.0586 | 0.0835 | 0.0404 | 0.1372 | 0.0768 | 0.0961 | 0.0510 |
| DisenCDR | 0.0825 | 0.0407 | 0.0584 | 0.0306 | 0.1094 | 0.0581 | 0.0843 | 0.0412 | 0.1390 | 0.0781 | 0.0987 | 0.0515 |
| ETL | 0.0851 | 0.0426 | 0.0597 | 0.0324 | 0.1125 | 0.0590 | 0.0863 | 0.0435 | 0.1406 | 0.0795 | 0.0998 | 0.0519 |
| VDEA | 0.0886 | 0.0450 | 0.0617 | 0.0345 | 0.1138 | 0.0603 | 0.0872 | 0.0446 | 0.1417 | 0.0810 | 0.1003 | 0.0524 |
| DURation | 0.0906 | 0.0472 | 0.0630 | 0.0369 | 0.1158 | 0.0626 | 0.0891 | 0.0454 | 0.1425 | 0.0817 | 0.1021 | 0.0532 |
| CL-DTCDR | 0.0924 | 0.0493 | 0.0636 | 0.0381 | 0.1175 | 0.0642 | 0.0914 | 0.0475 | 0.1442 | 0.0831 | 0.1053 | 0.0547 |
| MOTKD | 0.0935 | 0.0506 | 0.0664 | 0.0389 | 0.1192 | 0.0651 | 0.0923 | 0.0487 | 0.1464 | 0.0842 | 0.1078 | 0.0559 |
| **SIEOUG**-O | 0.0902 | 0.0486 | 0.0641 | 0.0375 | 0.1169 | 0.0637 | 0.0898 | 0.0468 | 0.1423 | 0.0814 | 0.1050 | 0.0538 |
| **SIEOUG**-M | 0.0949 | 0.0521 | 0.0675 | 0.0413 | 0.1231 | 0.0682 | 0.0958 | 0.0502 | 0.1507 | 0.0874 | 0.1126 | 0.0594 |
| **SIEOUG**-G | 0.0958 | 0.0530 | 0.0692 | 0.0427 | 0.1244 | 0.0689 | 0.0971 | 0.0508 | 0.1511 | 0.0865 | 0.1117 | 0.0586 |
| **SIEOUG** | **0.1031** | **0.0595** | **0.0763** | **0.0476** | **0.1296** | **0.0724** | **0.1007** | **0.0530** | **0.1555** | **0.0889** | **0.1158** | **0.0612** |

| Food & Kitchen | Food ($\mathcal{K}_u = 10\%$) | | Kitchen ($\mathcal{K}_u = 10\%$) | | Food ($\mathcal{K}_u = 50\%$) | | Kitchen ($\mathcal{K}_u = 50\%$) | | Food ($\mathcal{K}_u = 90\%$) | | Kitchen ($\mathcal{K}_u = 90\%$) | |
|---|---|---|---|---|---|---|---|---|---|---|---|---|
| | HR@10 | NDCG@10 | HR@10 | NDCG@10 | HR@10 | NDCG@10 | HR@10 | NDCG@10 | HR@10 | NDCG@10 | HR@10 | NDCG@10 |
| NeuMF | 0.0533 | 0.0119 | 0.0174 | 0.0102 | 0.1263 | 0.0381 | 0.0496 | 0.0215 | 0.1437 | 0.0658 | 0.0602 | 0.0331 |
| LightGCN | 0.0770 | 0.0221 | 0.0253 | 0.0184 | 0.1451 | 0.0469 | 0.0512 | 0.0293 | 0.1505 | 0.0716 | 0.0767 | 0.0446 |
| LATTICE | 0.0994 | 0.0338 | 0.0391 | 0.0243 | 0.1366 | 0.0556 | 0.0585 | 0.0304 | 0.1575 | 0.0772 | 0.0827 | 0.0508 |
| FREEDOM | 0.1045 | 0.0442 | 0.0456 | 0.0301 | 0.1396 | 0.0578 | 0.0640 | 0.0383 | 0.1617 | 0.0864 | 0.0859 | 0.0532 |
| LGM3Rec | 0.1078 | 0.0475 | 0.0471 | 0.0345 | 0.1421 | 0.0609 | 0.0673 | 0.0426 | 0.1644 | 0.0892 | 0.0887 | 0.0556 |
| DisenCDR | 0.0982 | 0.0384 | 0.0393 | 0.0294 | 0.1375 | 0.0601 | 0.0656 | 0.0410 | 0.1686 | 0.0907 | 0.0915 | 0.0563 |
| ETL | 0.1019 | 0.0403 | 0.0422 | 0.0317 | 0.1404 | 0.0618 | 0.0679 | 0.0423 | 0.1709 | 0.0924 | 0.0923 | 0.0574 |
| VDEA | 0.1026 | 0.0435 | 0.0504 | 0.0330 | 0.1419 | 0.0627 | 0.0705 | 0.0445 | 0.1712 | 0.0931 | 0.0928 | 0.0589 |
| DURation | 0.1071 | 0.0494 | 0.0525 | 0.0363 | 0.1437 | 0.0646 | 0.0723 | 0.0468 | 0.1730 | 0.0939 | 0.0942 | 0.0585 |
| CL-DTCDR | 0.1115 | 0.0537 | 0.0566 | 0.0384 | 0.1454 | 0.0679 | 0.0728 | 0.0481 | 0.1738 | 0.0945 | 0.0964 | 0.0593 |
| MOTKD | 0.1164 | 0.0596 | 0.0603 | 0.0417 | 0.1482 | 0.0685 | 0.0749 | 0.0506 | 0.1753 | 0.0968 | 0.0989 | 0.0617 |
| **SIEOUG**-O | 0.1141 | 0.0565 | 0.0588 | 0.0392 | 0.1463 | 0.0668 | 0.0701 | 0.0474 | 0.1707 | 0.0936 | 0.0941 | 0.0573 |
| **SIEOUG**-M | 0.1183 | 0.0602 | 0.0627 | 0.0425 | 0.1556 | 0.0754 | 0.0774 | 0.0539 | 0.1781 | 0.1012 | 0.1005 | 0.0641 |
| **SIEOUG**-G | 0.1202 | 0.0611 | 0.0649 | 0.0437 | 0.1570 | 0.0776 | 0.0788 | 0.0545 | 0.1774 | 0.0998 | 0.1001 | 0.0636 |
| **SIEOUG** | **0.1279** | **0.0663** | **0.0726** | **0.0498** | **0.1625** | **0.0833** | **0.0854** | **0.0586** | **0.1813** | **0.1031** | **0.1033** | **0.0659** |

different domains and thus limits the model potentials. (2) Conventional cross-domain recommendation models (e.g., **DisenCDR**) can provide better results than most of single-domain recommendation models (e.g., **LightGCN**) indicates the importance of leveraging useful knowledge across domains. However, these methods did not incorporate multi-modal information, resulting in less expressive item embeddings. (3) Some latest multi-modal cross-domain recommendation models (e.g., **MOTKD**) achieve much better results against other baselines. Nonetheless, **MOTKD** did not fully consider the guidance among these overlapped users for domain adaptation via optimal transport, and thus it may lead to ambiguous matching solutions and deteriorate the model performance. (4) Our proposed **SIEOUG** reaches state-of-the-art performance demonstrates the efficacy of similarity item exploration and overlapped user guidance. Moreover, **SIEOUG** can boost model performance even in cases where only relatively few users are overlapped (e.g., $\mathcal{K}_u = 10\%$) among different three tasks which is rather challenging.

Furthermore, we collect HR@10 and NDCG@10 of the runner-up model **MOTKD** and our proposed **SIEOUG** for Amazon Clothes and Amazon Sports with $\mathcal{K}_u = 90\%$ as shown in Fig.6. We can observe that **SIEOUG** reaches much better performance during the whole training epoch, showing the stability of **SIEOUG**.

## 4.4 Analysis (for RQ2 and RQ4)

**Ablation.** To study how does each module of **SIEOUG** contribute on the final performance, we compare **SIEOUG** with its several variants, including **SIEOUG**-O, **SIEOUG**-M, and **SIEOUG**-G. **SIEOUG**-O does not involve overlapped user guidance module which means it functions as a single-domain recommendation model with $\lambda = 0$. **SIEOUG**-M simply adopts the user contrastive loss among overlapped users. **SIEOUG**-G replaces GOUM with conventional fused Gromov-Wasserstein optimal transport without guidance among source and target users. The comparison results are shown in Table.1. From it, we can observe that: (1) **SIEOUG**-O outperforms

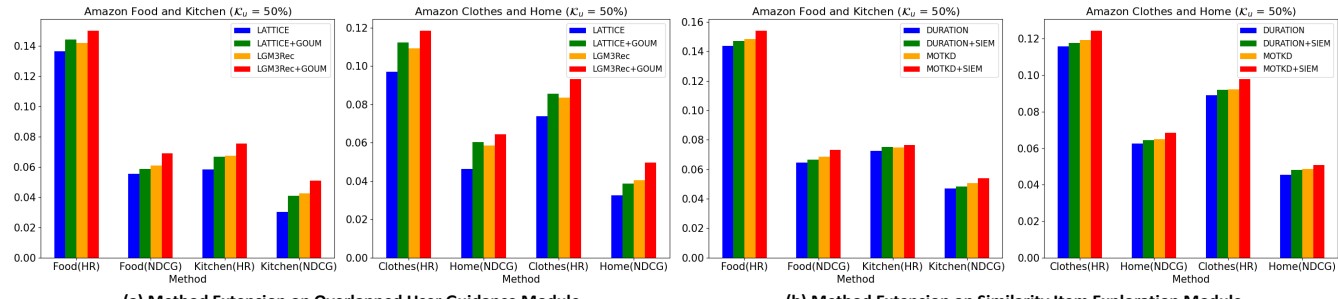

(a) Method Extension on Overlapped User Guidance Module
(b) Method Extension on Similarity Item Exploration Module

Figure 5: The experimental results on method extension.

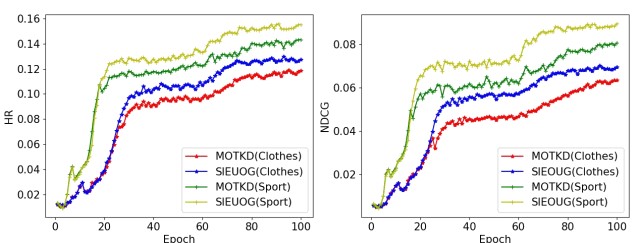

Figure 6: HR@10 and NDCG@10 of MOTKD and SIEOUG for Amazon Clothes and Amazon Sports with $\mathcal{K}_u = 90\%$.

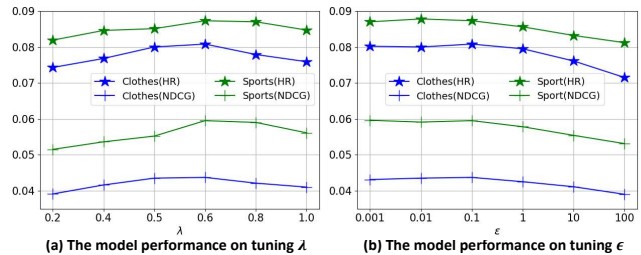

(a) The model performance on tuning $\lambda$
(b) The model performance on tuning $\epsilon$

Figure 7: The experimental results on hyper parameters.

other single-domain recommendation models (e.g., **FREEDOM**) indicates our proposed similarity item exploration module can better exploit multi-modal information for solving **CH1** within each domains. (2) By comparing the performance results of **SIEOUG** and **SIEOUG**-O, it shows that the results of **SIEOUG**-O are inferior to that of **SIEOUG** since it fails to transfer useful knowledge across domains via the overlapped users. (3) **SIEOUG**-M and **SIEOUG**-G achieve much better performance than **SIEOUG**-O and other CDR baselines (e.g., **MOTKD**). However, **SIEOUG**-M is limited by the number of overlapped users especially when $\mathcal{K}_u$ is much smaller. Meanwhile **SIEOUG**-G fails to consider the guidance among the overlapped users and thus it could lead to negative transfer. (4) **SIEOUG** achieves the better performance than **SIEOUG**-M, showing the proposed overlapped user guidance module can sufficiently share useful knowledge across domains for tackling **CH2**.

**Method Extension.** We further analyse the general extension of our proposed GOUM algorithm with guidance user contrastive loss in **SIEOUG**. Specifically, we aim to integrate GOUM into the single-domain multi-modal recommendation models **LATTICE** and **LGM3Rec**, resulting in **LATTICE**+GOUM and **LGM3Rec**+GOUM. Then we conduct the experiments on Amazon Food ↔ Amazon Kitchen and Amazon Clothes ↔ Amazon Home and report the

results in Fig.5(a)-(b). From this, we can observe that both **LATTICE**+GOUM and **LGM3Rec**+GOUM outperform the original models, indicating that the proposed method has strong generalization capabilities. Moreover, we analyse the extension of our proposed Similarity Item Exploration Module (SIEM) with item-item graph and hypergraph on **DURation** and **MOTKD** to construct **DURation**+SIEM and **MOTKD**+SIEM. The results are shown in Fig.5(c)-(d) and both **DURation**+SIEM and **MOTKD**+SIEM boost the model performance, showing SIEM can be also applied to other models.

**Effect of hyper-parameters.** We finally study the effects of hyper-parameters on model performance in Amazon Clothes ↔ Amazon Sport with user overlapped ratio $\mathcal{K}_u = 10\%$. Specifically, we vary $\lambda$ in range of $\{0.2, 0.4, 0.5, 0.6, 0.8, 1.0\}$ and report the results of HR@10 and NDCG@10 in Fig.7(a). We can observe that when $\lambda$ is smaller (e.g., $\lambda = 0.2$), the guidance user contrastive loss is less effective for knowledge transfer. Meanwhile when $\lambda$ is larger (e.g., $\lambda = 1$), it may negatively impact the contrastive collaborative filtering loss in each domain, leading to deteriorated results. Therefore we set $\lambda = 0.6$ empirically. Moreover, we conduct the experiments on varying $\epsilon$ in range of $\{0.001, 0.01, 0.1, 1, 10, 100\}$ on GOUM and report the results of HR@10 and NDCG@10 in Fig.7(b). We can conclude that the model can obtain better performance when $\epsilon$ is smaller which aligns with the limitation results in Eq.(20). However, when $\epsilon$ increases, the approximate calculations may result in inaccurate matching solutions, compromising the overall accuracy. Therefore we set $\epsilon = 0.1$ empirically for **SIEOUG**.

## 5 CONCLUSION

In this paper, we Joint Similarity Item Exploration and Overlapped User Guidance model (**SIEOUG**) for solving the Multi-Modal Cross-Domain Recommendation (MMCDR) problem, which includes the *similarity item exploration module*, *user-item collaborative filtering module*, and the *overlapped user guidance module*. Similarity item exploration module constructs item-item graph and hypergraph via proposed RISGF and SISHE algorithm from pair-wise and group-wise perspectives. User-item collaborative filtering module further aggregates multi-modal information for user-item modeling within each domain. Overlapped user guidance module realizes the knowledge transfer among the source and target users via GOUM with optimal matching accordingly. By incorporating these three modules, **SIEOUG** can not only fully exploit and utilize multi-modal item information, but also sharing useful knowledge to tackle data sparsity problem. We also conduct extensive experiments on Amazon datasets to demonstrate the superior performance of our proposed **SIEOUG** on several tasks under different overlapped user ratio.

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

# Appendix

## A SPARSITY ITEM SIMILARITY HYPERGRAPH EXPLORATION

The algorithm of Sparsity Item Similarity Hypergraph Exploration (SISHE) is provided in Alg.1.

---

**Algorithm 1** The procedure scheme of Sparsity Item Similarity Hypergraph Exploration.

---

1: **Input:** $A^d$: pair-wise item similarity graph in domain $d$; $K$: Number of clusters for constructing group-wise item hypergraph.
2: Initialize the clustering matrix as $(\gamma^d)_{ij}^{(0)} = \frac{1}{K}$.
3: **for** $l = 0$ to $T - 1$ **do**
4:     Calculate the matrix $(-A^d(\gamma^d)^{(l)} I_K) = \mathcal{Y}^{(l)}$.
5:     Update the value of Lagrange multiplier $f$ via:

$$f_i = \begin{cases} \mathcal{G}_o^{(l)} + \frac{\eta - \phi(\mathcal{G}_o^{(l)})}{\phi(\mathcal{G}_{o+1}^{(l)}) - \phi(\mathcal{G}_o^{(l)})}(\mathcal{G}_{o+1}^{(l)} - \mathcal{G}_o^{(l)}), & \phi(\mathcal{G}_o^{(l)}) \le \eta < \phi(\mathcal{G}_{o+1}^{(l)}) \\ \frac{1}{K}\left(\eta + \sum_{j=1}^{K} \mathcal{G}_j^{(l)}\right), & \eta \ge \phi(\mathcal{G}_K^{(l)}) \end{cases}$$

6:     Update the value of Lagrange multiplier $g$ via:

$$g_j = \begin{cases} \mathcal{F}_o^{(l)} + \frac{\eta\frac{N}{K} - \psi(\mathcal{F}_o^{(l)})}{\psi(\mathcal{F}_{o+1}^{(l)}) - \psi(\mathcal{F}_o^{(l)})}(\mathcal{F}_{o+1}^{(l)} - \mathcal{F}_o^{(l)}), & \psi(\mathcal{F}_o^{(l)}) \le \eta\frac{N}{K} < \psi(\mathcal{F}_{o+1}^{(l)}) \\ \frac{1}{N}\left(\eta\frac{N}{K} + \sum_{i=1}^{N} \mathcal{F}_j^{(l)}\right), & \eta\frac{N}{K} \ge \psi(\mathcal{F}_K^{(l)}) \end{cases}$$

7:     Update the clustering matrix via:

$$(\gamma^d)_{ij}^{(l+1)} = \left[\frac{f_i + g_j - \mathcal{Y}_{ij}^{(l)}}{\eta}\right]_+$$

8: **end for**
9: **Return:** The clustering matrix $\gamma^d$.

---

## B GUIDANCE-BASED OPTIMAL USER MATCHING

The algorithm of Guidance-based Optimal User Matching (GOUM) is provided in Alg.2.

**Table 2: Statistics on Douban and Amazon datasets.**

| Datasets | | Users | Items | Ratings | Sparsity |
|---|---|---|---|---|---|
| Task 1 | Amazon Clothes | 9,202 | 10,652 | 83,130 | 99.92% |
| | Amazon Sports | | 10,219 | 84,013 | 99.91% |
| Task 2 | Amazon Clothes | 6,548 | 7,266 | 55,789 | 99.88% |
| | Amazon Home | | 8,418 | 71,171 | 99.87% |
| Task 3 | Amazon Food | 9,815 | 9,949 | 135,634 | 99.86% |
| | Amazon Kitchen | | 14,680 | 154,152 | 99.89% |

## C EMPIRICAL STUDY

**Datasets and Tasks.** We conduct extensive experiments on commonly used real-world **Amazon** datasets. The **Amazon** dataset [42] has five domains, i.e., Clothing Shoes and Jewelry (Clothes), Sports and Outdoors (Sports), Tools and Home Improvement (Home), Grocery and Gourmet Food (Food), and Home and Kitchen (Kitchen). For each dataset, we binarize the ratings higher or equal to 4 as 1 and the rest as 0 following [24]. Then we filter the users and items

---

**Algorithm 2** The procedure scheme of Guidance-based Optimal User Matching.

---

1: **Input:** $\epsilon$: The hyper parameter; $C$: The cost matrix; $\mathcal{M}$: The guidance mask matrix.
2: Adding the mask into the cost matrix via $C_{ij} \odot \mathcal{M}_{ij} = Q_{ij}$.
3: Adopting FISTA($\mathcal{J}_{KP}$) to optimize the optimization problem:

$$\min_{\widehat{\omega}} \mathcal{J}_{KP} = \sum_{j=1}^{N}\left\{\epsilon \log\left[\sum_{i=1}^{N} \exp\left(\frac{\widehat{\omega}_i - Q_{ij}}{\epsilon}\right)\right]\right\} - \sum_{i=1}^{N} \widehat{\omega}_i$$

4: Obtain the optimal solution on $\widehat{\omega}$ and $\widehat{\kappa}$ via:

$$\widehat{\kappa}_j = -\epsilon \log\left[\sum_{i=1}^{N} \exp\left(\frac{\widehat{\omega}_i - Q_{ij}}{\epsilon}\right)\right]$$

5: Obtain the user-user matching results:

$$\pi_{ij} = \begin{cases} 1, & \arg\min_j \left[Q_{ij} - \widehat{\omega}_i - \widehat{\kappa}_j\right] \\ 0, & \text{otherwise} \end{cases}$$

6: **Function:** FISTA($\mathcal{J}_{KP}$):
7: Initialize $\theta_0 = 1$, $t' = 0$, and $\widehat{\omega}^{(0)} = (0, 0, \cdots, 0)$.
8: **repeat**
9:     Calculate the gradient of ($\mathcal{J}_{KP}$) as:

$$\mathcal{G}(\widehat{\omega}_i^{(t)}) = \frac{\partial \mathcal{J}_{KP}}{\partial \widehat{\omega}_i^{(t)}} = \sum_{j=1}^{N}\left[\frac{\exp((\widehat{\omega}_i^{(t)} - Q_{ij})/\epsilon)}{\sum_{k=1}^{M} \exp((\widehat{\omega}_k^{(t)} - Q_{kj})/\epsilon)}\right] - 1$$

10:     Update $\widetilde{\omega}^{(t)} = \widehat{\omega}^{(t)} - \beta_s \mathcal{G}(\widehat{\omega}^{(t)})$ where $\beta_s$ denotes the step size that learned from the Armijo line search method.
11:     Update $\theta_{t'+1} = \frac{1}{2}(1 + \sqrt{1 + 4\theta_{t'}^2})$
12:     Update $\widehat{\omega}^{(t)} \leftarrow \widehat{\omega}^{(t)} + \frac{\theta_{t'} - 1}{\theta_{t'+1}}(\widetilde{\omega}^{(t)} - \widehat{\omega}^{(t)})$.
13:     Update $t' = t' + 1$.
14: **until** Converge.
15: **Return:** The user-user matching matrix $\pi$.

---

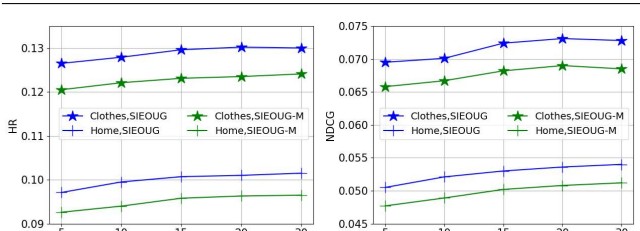

**Figure 8: The experimental results on number of clusters $K$.**

with less than 10 interactions. The detailed datasets statistics after prepossessing are shown in Table 2.

**Effect of hyper-parameters.** We also vary the number of cluster $K$ in range of $K \in \{5, 10, 15, 20, 30\}$ in Amazon Clothes and Home with $\mathcal{K}_u = 50\%$ and report the results (HR and NDCG) in Fig.8. We observe that as the number of clusters increases, performance improves significantly. However, when the number of clusters reaches a certain value as $K = 15$, its growth effect slows down. Meanwhile, a large number of clusters can also lead to increased computational burdens. Therefore we set $K = 15$ empirically.