# OpenReview forum: "Joint Similarity Item Exploration and Overlapped User Guidance for Multi-Modal Cross-Domain Recommendation"
_ACM.org/TheWebConf/2025/Conference — WWW 2025 Poster_

### Official Review · Reviewer_MEmh · 2024-11-29

**Novelty:** 4
**Technical Quality:** 3

**Review:**

# Review:
This work proposed a method called SIEOUG to solve the multimodal cross-domain recommendation (MMCDR) problem by jointing item similarity and overlapped user information. Specifically, SIEOUG combines pair and group-wise item similarity to explore items' multimodal features and utilizes the overlapped users to guide knowledge transfer across different domains.

## Pros:
- Technically, it is reasonable to aggregate pair-wise and group-wise item similarity in the recommendation task.
- The paper is relatively understandable and provides detailed modeling and optimization processes.
- The experiments showed the effectiveness of the presented approach under multiple MMCDR settings.

## Cons:
- The rationale for some procedures should be further emphasized, and the illustration for the GOUM module is unclear.
- The ablation study is insufficient without fully demonstrating the usefulness of each module.
- There are several typos in the paper. In equation (23), the symbol in the bracket might be '-,' not '+.' Sub-captions (c) and (d) are missing in Figure 5.

**Questions:**

- In equation (6), why does the SISHE module constrain the summation of each column to N/K? Does it mean that each cluster contains the same quantity of items？
- I am also interested in whether using soft similarity in equation (2) rather than 0/1 hard similarity is feasible.
- Could the authors provide a detailed explanation of Figure 4, which would help readers further understand the GOUM module?
- I would like to see more ablations about the SISHE and collaborative filtering modules. Could the authors compare the performance of single pair-wise or group-wise item similarity?
- Moreover, it would be better to provide more comparisons with the latest methods proposed in 2023 and 2024.

**Reviewer Confidence:**

3: The reviewer is confident but not certain that the evaluation is correct

**Scope:**

3: The work is somewhat relevant to the Web and to the track, and is of narrow interest to a sub-community

---

### Official Review · Reviewer_KJMH · 2024-12-01

**Novelty:** 3
**Technical Quality:** 4

**Review:**

The authors present a novel approach, Joint Similarity Item Exploration and Overlapped User Guidance (SIEOUG), for the multimodal cross-domain recommendation, which can be crucial in recent recommendation systems. The empirical results show superior performance. By utilizing information from multimodal item features and matching non-overlapped users, the proposed approach could lead to more accurate and robust recommendations. However, there are some concerns with the paper.

- Although the authors paired several domains, they solely rely on single datasets (i.e., Amazon). It would be better to experiment with other datasets.
- The authors conducted analysis on matching non-overlapped users and overlapped user guidance (GOUM). However, there is a lack of analysis on RISGF and SISHE.
- It seems some information is missing or wrong, so the authors would better check out the paper (i.e., explanation of notation '$\langle \cdot, \cdot \rangle$', subfigure indexing in Figure 5 does not match the text).

**Questions:**

- Equation 17 shows that $\sum^N_{j=1} \pi_{ij} = 1$ and $\sum^N_{i=1} \pi_{ij} = 1$. However, it is natural to think the user pools do not match in distinct domains in real-world scenarios. How can you adjust the proposed method in this case?
- Would it be effective in the case where the user pools are not overlapped at all?
- Can you provide additional information about computational cost or memory usage to baseline methods?
- Addressing the concerns above.

**Reviewer Confidence:**

3: The reviewer is confident but not certain that the evaluation is correct

**Scope:**

4: The work is relevant to the Web and to the track, and is of broad interest to the community

---

### Official Review · Reviewer_S1A5 · 2024-12-02

**Novelty:** 5
**Technical Quality:** 5

**Review:**

The paper proposes a novel framework, SIEOUG (Joint Similarity Item Exploration and Overlapped User Guidance), to address the challenges of Multi-Modal Cross-Domain Recommendation (MMCDR). The authors identify the major limitations of existing MMCDR methods, such as their inability to fully leverage multi-modal item information and their reliance on overlapped users for knowledge transfer, which significantly hampers performance when user overlap is sparse. To overcome these issues, SIEOUG introduces three key modules: Similarity Item Exploration Module (to model pair-wise and group-wise item similarities), User-Item Collaborative Filtering Module (to aggregate multi-modal embeddings for intra-domain recommendation), and Overlapped User Guidance Module (to utilize optimal user matching for knowledge transfer across domains).

**Questions:**

1. The model's core components, such as RISGF and SISHE, involve complex optimization processes and require solving multiple Optimal Transport (OT) problems. This raises concerns about the method's convergence behavior. Could the authors provide a comparison of training time between SIEOUG and other SOTA methods when achieving the results reported in Table 1? Such a comparison would help clarify the computational efficiency and practicality of the proposed method.
2. The proposed Guidance-based Optimal User Matching (GOUM) method appears to share some similarities with InfoOT[1]  in terms of its objectives. Could the authors elaborate on the similarities and differences between these two approaches?

[1] Chuang, Ching-Yao, Stefanie Jegelka, and David Alvarez-Melis. "InfoOT: Information maximizing optimal transport." ICML, 2023.

**Reviewer Confidence:**

2: The reviewer is willing to defend the evaluation, but it is likely that the reviewer did not understand parts of the paper

**Scope:**

4: The work is relevant to the Web and to the track, and is of broad interest to the community

---

### Official Review · Reviewer_a9w6 · 2024-12-02

**Novelty:** 5
**Technical Quality:** 5

**Review:**

This paper focuses on the multi-modal cross domain recommendation (MMCDR) and proposes a framework (Joint Similarity Item Exploration and Overlapped User Guidance, SIEOUG). Specifically, MMCDR aims to alleviate the data sparsity problem by leveraging multi-modal information and knowledge transfer across domains, even when user overlap is minimal. And the authors introduce three core modules, Similarity Item Exploration Module, User-Item Collaborative Filtering Module, and Overlapped User Guidance Module. Finally, the extensive experiments on the real-world dataset demonstrate that SIEOUG significantly outperforms the baselines.

Strengths:
1. In this paper, the similarity item exploration module reduces irrelevant noise and captures both pair-wise and group-wise item relationships, improving the quality of item embeddings.
2. SIEOUG performs effectively even when user overlap is sparse, unlike traditional CDR models that depend heavily on overlapped users.
3. The framework demonstrates stability and effectiveness across varying levels of user overlap.

Weaknesses:
1. Although the overlapped user guidance module is effective, the framework does not thoroughly explore methods for directly modeling non-overlapped users.
2. The paper lacks detailed visualizations or interpretability analyses of the learned embeddings or item clusters, which could provide deeper insights into the effectiveness of the similarity item exploration module.
3. Incorporating multi-modal information adds complexity, which could reduce the practicality of SIEOUG in resource-limited environments or real-time applications.

**Questions:**

Please see the weaknesses above.

**Reviewer Confidence:**

3: The reviewer is confident but not certain that the evaluation is correct

**Scope:**

4: The work is relevant to the Web and to the track, and is of broad interest to the community